# Probiotics counteract the expression of hepatic profibrotic genes via the attenuation of TGF-β/SMAD signaling and autophagy in hepatic stellate cells

**Paulraj Kanmani[1,2], Hojun Kim[1]***

**1** Department of Korean Medicine, Dongguk University, Goyang, Republic of Korea, **2** Department of Medicine, University of Illinois, Chicago, Illinois, United States of America

* kimklar@dongguk.ac.kr

**Data Availability Statement:** All relevant data are within the paper and its Supporting Information files.

## Abstract

Hepatic fibrosis is caused by the increased accumulation and improper degradation of extracellular matrix (ECM) proteins in the liver. Hepatic stellate cells (HSCs) activation is a key process in initiating hepatic fibrosis and can be ameliorated by the administration of probiotic strains. This study hypothesized that LAB strains (*Lactiplantibacillus plantarum*, *Lactobacillus brevis*, and *Weissella cibaria*) might attenuate pro-fibrogenic cytokine TGF-β mediated HSCs activation and induce collagen deposition, expression of other fibrogenic/inflammatory markers, autophagy, and apoptotic processes *in vitro*. Few studies have evaluated the probiotic effects against fibrogenesis *in vitro*. In this study, TGF-β exposure increased collagen deposition in LX-2 cells, but this increase was diminished when the cells were pretreated with LAB strains before TGF-β stimulation. TGF-β not only increased collagen deposition, but it also significantly upregulated the mRNA levels of Col1A1, alpha-smooth muscle actin (α-SMA), matrix metalloproteinases-2 (MMP-2), IL-6, CXCL-8, CCL2, and IL-1β in LX-2 cells. Pretreatment of the cells with LAB strains counteracted the TGF-β-induced pro-fibrogenic and inflammatory markers by modulating SMAD-dependent and SMAD-independent TGF-β signaling. In addition, LX-2 cells exposed to TGF-β induced the autophagic and apoptotic associated proteins that were also positively regulated by the LAB strains. These findings suggest that LAB can attenuate TGF-β signaling that is associated with liver fibrogenesis.

## Introduction

The broad functional properties of probiotics have led to their increased use in several fields, including clinical and medical fields. Probiotics are live microorganisms, and their treatment ameliorates intestinal disease [1], renal complications [2], brain [3], lung [4], and cardiovascular diseases [5]. Apart from these diseases, probiotics have also been tested *in vivo* against various liver disease models, including alcoholic [6], nonalcoholic fatty liver disease [7], nonalcoholic steatohepatitis [8], and hepatocellular carcinoma [9]. These studies revealed the potent protective effects of probiotics against hepatic diseases and different mechanisms

**Funding:** This study was supported by a National Research Foundation (NRF) of Korea grant funded by the Korean Government to PK (NRF-2019R11A1A01058795), and also was supported by a grant of the Korea Health Technology R&D Project through the Korea Health Industry Development Institute (KHIDI), funded by the Ministry of Health &Welfare, Republic of Korea (grant number: HF20C0020) to HK (NRF-2019R1A2B5B01070365) as well as by the Korean Research Fellowship (KRF) program of the NRF (2016H1D3A1937971). The funders did not play any role in the study design, data collection, and analysis, decision to publish, or preparation of the manuscript.

**Competing interests:** The authors declare that there is no conflict of interest

behind the beneficial effects of probiotics. On the other hand, the available reports failed to provide detailed mechanisms against liver fibrosis. Furthermore, no studies have examined the effects of probiotics against hepatic fibrosis *in vitro*. Hepatic fibrosis is caused by the increased accumulation of extracellular matrix (ECM) proteins, including collagen in the liver [10]. It is a major global health problem, and more than 100 million people are suffering from this condition worldwide. Unfortunately, there is no adequate therapy for liver fibrosis that can lead to cirrhosis and hepatocellular carcinoma. Thus far, probiotics with potent protective effects would have an advantage in treating this disease.

Liver fibrosis (LF) is caused by different factors, including alcoholic and nonalcoholic diseases, viral hepatitis, and autoimmune diseases [11, 12]. These factors induce the generation and proliferation of contractile myofibroblasts (MFB) subsequent to the activation of quiescent hepatic stellate cells [13]. Both HSCs/MFB are key players involved in the excessive production and synthesis of ECM, resulting in LF formation. The activation of these cells induces matrix synthesis and the production and expression of several proinflammatory cytokines and growth factors that participate in liver injury and inflammation [14]. In addition, liver pathological conditions induce other immune cell types, such as kupffer cells that produce an array of pro-fibrogenic and proinflammatory cytokines, which stimulate and activate HSCs, resulting in the development of LF and hepatocellular carcinoma (HCC) [15]. Among the cytokines, transforming growth factor-β (TGF-β) is a prime pro-fibrogenic cytokine that plays a pivotal role in the progression and development of LF [15]. The elevation of TGF-β, as a consecutive liver injury or disease, activates HSCs, promotes the differentiation of MFB, and induces the excessive production of ECM in the liver [16].

TGF-β isoforms trigger the intracellular fibrogenic signaling pathway by interacting with transmembrane TGF-β receptor II (TβRII). The recruitment and activation of TβRI kinase by TGF-β/TβRII phosphorylates and activates the SMAD2/SMAD3 transcription factors, which form a complex with SMAD4. This SMAD-containing heteromeric complex then translocates to the nucleus and regulates the expression of the target genes [17]. In response to TGF-β, HSCs show an increase in the expression of collagen (COL1A1, COL3A1), matrix metalloproteinases (MMPs), and alpha-smooth muscle actin (α-SMA), which are the major indicators of LF [15, 18]. TGF-β also mediates the non-SMAD signaling pathway, including MAPKs, mTOR, and others. However, their contributions to LF have not been investigated in detail [19]. In addition, the TLR4/LPS pathway actively contributes to the activation of TGF-β signaling in HSCs and LF [15]. Pretreatment of HSCs with LPS increases the response of HSCs to TGF-β by down-regulating bone morphogenic protein and activin membrane-bound inhibitor (BAMBI). Furthermore, the LPS treatment increases the generation of NF-κBp50 that binds with HDAC1 to repress BAMBI, which induces the sensitivity to TGF-β signaling in human HSCs [20].

Because HSCs is a potent therapeutic target in the pathogenesis of LF, several studies have used HSCs to examine the mechanisms linked to the inhibition and activation of HSCs. For example, a recent study reported that the treatment of LX-2 cells with TGF-β increased the level of collagen, α-SMA, fibronectin, CTGF, MMP2, MMP-9, TIMP1, and TIMP2 via the suppression of STAT3 phosphorylation [21]. Therefore, in this study, a human HSCs (LX-2) was used to evaluate the anti-fibrogenic effects of probiotic strains against TGF-β and study the molecular mechanisms behind the immunomodulatory and anti-fibrogenic activity of the selected probiotic strains.

## Materials and methods

### Cell culture

Prof. Scott Friedman, Mount Sinai Medical School, NY, kindly provided the human hepatic stellate cell line (LX-2). The cells were cultured in Dulbecco's modified Eagle's Medium

(DMEM, GIBCO, USA) supplemented with 10% or 1% of FBS and 1% penicillin/streptomycin at 37°C, in a humidified atmosphere containing 5% $CO_2$.

## Lactic acid bacteria

Lactic acid bacteria (LAB), such as *Lactiplantibacillus plantarum* DU1, *Lactobacillus brevis*, and *Weissella cibaria* DU1, were cultured in MRS broth at 37°C. After 19 h incubation, the cells were centrifuged, washed with DPBS, and suspended in PBS at the appropriate concentration [22]. The cells were then heat-killed by incubating at 70°C for 1h and stored at −80°C for further experiments. The cytotoxicity of the LAB strains (S1 Fig) in the LX-2 cells was analyzed using a cell viability, proliferation, and cytotoxicity assay kit (EZ-CYTOX, DOGEN Bio co. Ltd).

## *In vitro* analysis of collagen deposition

The LX-2 cells were seeded ($3 \times 10^4$ cells/well) in 12 well ($3.80$ cm$^2$) type I collagen-coated plates (SPL Life Science Co. Ltd, Korea) and incubated at 37°C in a humidified atmosphere containing 5% $CO_2$. After cells reached 70 to 80% confluence, they were serum-starved for 48 h in DMEM without FBS, followed by pre-stimulation with LABs ($5 \times 10^7$ cells/ml) for 48 h and washed twice with DPBS. The cells were then post-stimulated with 5 ng/ml of TGF-β (R&D systems, Minneapolis, MA, USA) for 12 h and 24 h. The level of collagen deposition in the cells was analyzed by picro-Sirius red staining.

## Picro-Sirius red staining

The treated LX cells were washed twice with PBS and fixed with 10% formalin at room temperature (RT). After 10–15 minutes of fixation, the formalin was removed, and the cells were washed with PBS and incubated with the picro-Sirius red solution (Picro Sirus Red stain kit, Abcam, MA, USA) at RT for 1 h. The cells were then washed with an acetic acid solution, PBS and observed under a fluorescence microscope (Leica DMI 6000B, Wetzlar, Germany). After taking the microscopic images, a 0.1N NaOH solution was added to each dry well and subsequently incubated at RT for 30 minutes. The slides were read at 540 nm using a microplate reader (SpectraMax Plus 384, San Jose, CA, USA) to quantify the amount of collagen deposited in the cells.

## Anti-fibrogenic activity of the LAB strains

The LX-2 cells were cultured ($3 \times 10^4$ cells/well) in 12 well ($3.80$ cm$^2$) type I collagen-coated plates at 37°C, 5% $CO_2$ for 4–5 days. The cells were stimulated with the LAB strains ($5 \times 10^7$ cells/ml) and incubated at 37°C, 5% $CO_2$. After 48 h, the cells were washed twice with DPBS and stimulated with TGF-β (5 ng/ml) for 24 h. The cells were then washed with DPBS and collected by adding 500 μl of TRIzol to each well. The collected samples were stored at −80°C to extract the RNA. The expression of the HSCs activation makers (LOX and LOXL-2), profibrogenic markers (α-SMA, Col1A1, MMP-2, ICAM-1, TIMP-1, and TIMP-2), and proinflammatory cytokines (IL-6, CXCL-8, CCL2, IL-1β, TNF-α, and TRAF6) were analyzed by RT-PCR, as mentioned below.

## Quantitative real-time polymerase chain reaction (qRT-PCR)

The total RNA was extracted by adding TRIzol reagent (Invitrogen) according to the method described by Kanmani and Kim [22]. The purity and quantity of the RNA were analyzed using the Nanodrop method (NanDrop Technologies, USA). The cDNA was synthesized using a

Thermal cycler (BIORAD, Hercules, CA, USA). RT-PCR was performed using a 7300 real-time PCR system (Roche Applied Science, Indianapolis, ID, USA) with SYBR green and target primers (S1 Table). A 20 μl volume of the reaction mixture containing 1 μl of cDNA, and 19 μl of the master mix including SYBR green, and forward and reverse primers (1pmol/μl), was used. Amplification was performed at 50˚C for 5 min; followed by 95˚C for 5 min. This was followed by 40 cycles at 95˚C for 15s, at 63˚C for 30s, and at 72˚C for 30s. The β actin was used as an internal control.

## Analysis of TGF-β induced pro-fibrogenic markers and SMAD proteins in LX-2 cells

The LX-2 cells were seeded ($1.8 \times 10^5$ cells/dish) in dishes (60 mm) and incubated at 37˚C in a humidified atmosphere containing 5% $CO_2$ for 4–5 days. The cells were then stimulated with *L. plantarum* DU1, *W. cibaria* DU1, and *L. brevis* ($5 \times 10^7$ cells/ml) at 37˚C, 5% $CO_2$ for 48 h, after which the cells were washed with DPBS and post-stimulated with TGF-β (5 ng/ml). After 120 minutes of TGF-β stimulation, the cells were washed twice with DPBS, and 200 μl of Cell-Lytic M cell lysis reagent (Sigma-Aldrich, St. Louis, MO, USA) was added to each well to lyse the cells. The lysed samples were collected and stored at −80˚C to analyze the protein levels of α-SMA, Col1A1, SMAD2, and SMAD7 by western blot, as mentioned below.

## Analysis of autophagy/apoptosis and MAPKs/NF-κB induction LX-2 cells

The LX-2 cells ($1.8 \times 10^5$ cells/dish) were cultured in dishes (60 mm) at 37˚C in an atmosphere containing 5% $CO_2$ for 4–5 days. The cells were stimulated with the LAB strains ($5 \times 10^7$ cells/ml) for 48 h, and then post-stimulated with TGF-β (5 ng/ml) for 120 minutes. The cells were collected by lysing the cells with CellLytic M cell lysis reagent and stored at −80˚C. The autophagy (ATG5, p62, LC3BI/II, p-AKT and p-mTOR), apoptosis (Bcl2, Bax, Caspase-3, MAPKs (p-p38 and p-ERK), and NF-κB (p-IκBα) associated proteins were analyzed by western blot, as mentioned below.

## Western blot analysis

The concentration of protein in the lysed sample was analyzed using a bicinchoninic acid assay kit (Thermo Scientific, Pierce, Rockford, IL) after heating at 95˚C for 5 min. First, the desired volume of samples was loaded in 10% SDS-polyacrylamide gels and run at a constant voltage. The separated proteins were then transferred to a nitrocellulose membrane (Trans-Blot Turbo$^{TM}$, BIO-RAD). Subsequently, the transferred membrane was cut at the desired part and incubated with a blocking buffer before incubation with the desired antibodies.

The level of p-SMAD2, SMAD7, ATG5, p62, p-AKT, p-mTOR, Bcl2, Bax, caspase-3, β-actin (Santa Cruz Biotechnology (Dallas, TX, USA), LC3BI/II, p-p38, p-ERK, α-SMA, Col1A1, and p-IκBα (Cell Signaling Technology, Beverly, MA, USA) were evaluated by incubating (4˚C) the membrane overnight with the appropriate primary antibodies at a 1:1000 dilution of their original antibodies. The goat anti-rabbit and anti-mouse IgG-HRP polyclonal antibodies (AbFrontier, Seoul, South Korea) were used as the secondary antibodies. The optical protein bands were detected by adding a mixture (1:1 ratio) of a western blot detection solution A and B (SUPEX, Neonex Co., Ltd, Postech, South Korea), after which the area of the densitogram peak was estimated using the Image J software (National Institute of Health, Bethesda, MD, USA).

## Statistical analysis

Statistical analyses were performed using the SPSS software package (SPSS 12.0, SPSS Inc., Chicago, IL, USA). One-way analysis of variance (ANOVA) was performed, and the

significance of each mean value was determined using a Tukey's multiple range test with significant levels of p<0.05.

## Results

### Effect of LAB on collagen deposition in LX-2 cells

This study first evaluated whether the LAB strains reduce the TGF-β induced collagen deposition *in vitro*. The LX-2 cells were pre-incubated with the LABs and then post-stimulated with TGF-β for 24 h. Stimulation of LX-2 cells with TGF-β increased collagen deposition at both 12 and 24 h (Fig 1A and 1B). The TGF-β-induced collagen deposition was reduced when the LX-2 cells pre-stimulated with the LAB strains. The reduction of collagen by each LAB strain showed a similar pattern at both stimulation hours. On the other hand, there were some noticeable differences between both stimulation hours. *L. plantarum* and *L. brevis* showed a better reduction in collagen deposition than TGF-β in 24 h. *W. cibaria* also reduced the level of collagen, but it was higher than the other LAB strains and medium alone.

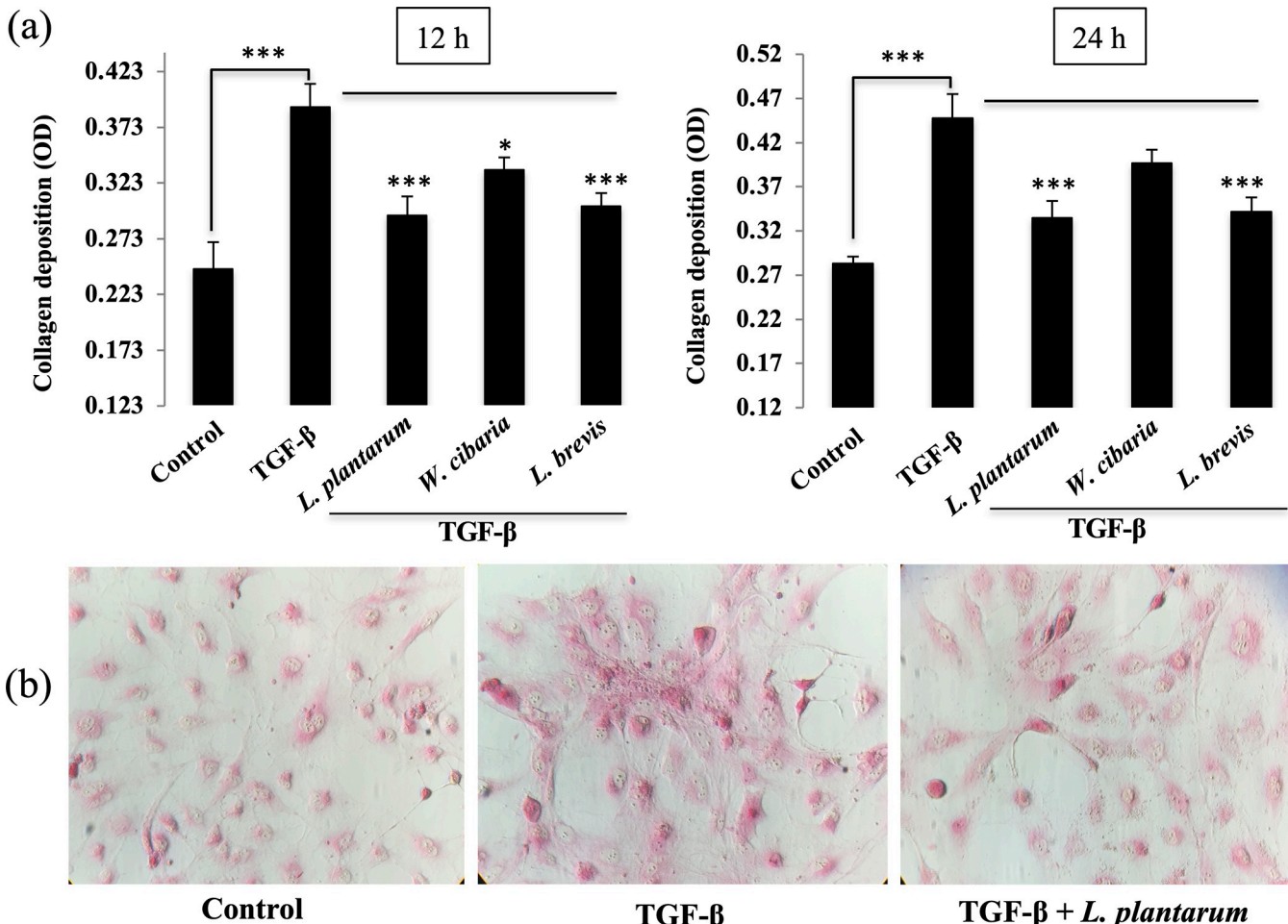

**Fig 1. LAB reduces the accumulation of collagen in LX-2 cells in response to TGF-β.** (A) The amount of collagen deposition in LX-2 cells was determined after 48 h incubation with LAB strains and with TGF-β for 12 h and 24 h. Two independent experiments (n = 3) were performed, and the average values (mean + S.D) are shown. *p <0.05, ** p <0.01, ***p <0.001. (B) Representative images (40 x) of collagen deposition for medium, TGF-β, and *L. plantarum*+TGF-β are showed.

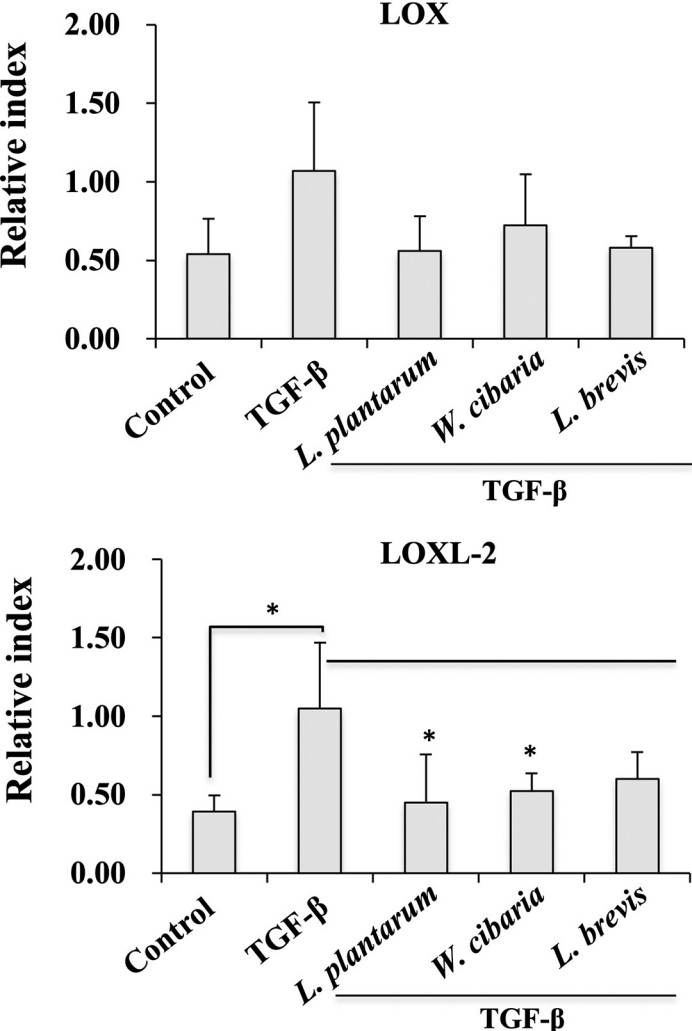

**Fig 2. LAB decreased the TGF-β-induced HSCs-activation markers *in vitro*.** Relative mRNA levels of the cell activation markers, such as LOX and LOXL-2 in LX-2 cells after stimulation with the LAB strains and TGF-β. LX-2 cells, either with the medium or TGF-β alone was used as a negative and positive control. Two independent experiments (n = 3) were performed, and the average values (mean + S.D) are shown. $^*p$ <0.05, $^{**}$ $p$ <0.01, $^{***}p$ <0.001.

## LAB reduces HSCs activation and pro-fibrogenic markers expression *in vitro*

The LX-2 cells were pretreated with LABs, followed by TGF-β stimulation for 24 h to determine if LAB decreases TGF-β induced cell activation and pro-fibrogenic markers expressions. The results are shown in Fig 2. Incubation of the cells with TGF-β tended to upregulate the expression of the HSCs activation markers, such as LOX and LOXL-2. The augmentation of the LOX and LOXL-2 levels upon TGF-β stimulation were reduced significantly by the LAB strains in LX-2 cells. *L. plantarum* and *L. brevis* exhibited a better reduction in the mRNA levels of the HSCs activation markers than *W. cibaria*. In addition, stimulation of LX-2 cells with TGF-β increased the expression of the pro-fibrogenic markers, such as α-SMA, Col1A1, MMP-2, ICAM-1, and TIMP-2. Pre-incubating the cells with the LAB strains before TGF-β

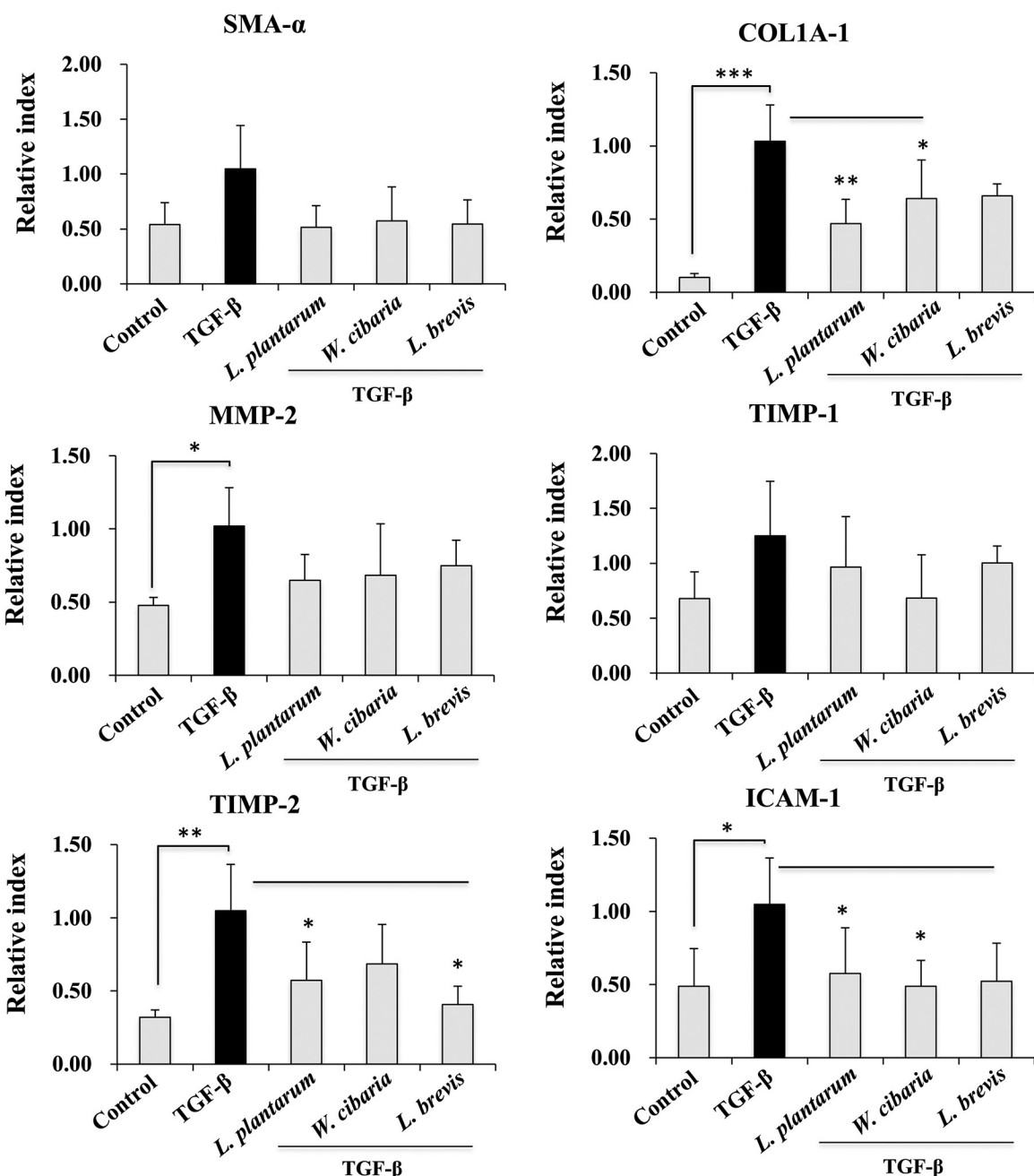

**Fig 3. Effect of LAB on TGF-β induced pro-fibrogenic markers expression in LX-2 cells.** Confluence LX-2 cells were treated with LAB for 48 h, then stimulated with TGF-β for 24 h. The mRNA levels of α-SMA, Col1A1, MMP-2, ICAM-1, TIMP-1, and TIMP2 were determined by RT-PCR. LX-2 cells, either with medium or TGF-β alone, were used as a negative and positive control. Two independent experiments (n = 3) were performed, and the average values (mean + S.D) are shown. $^*p$ <0.05, $^{**}$ $p$ <0.01, $^{***}p$ <0.001.

modulated these up-regulations (Fig 3). The mRNA level of α-SMA, Col1A1, MMP-2, ICAM-1, and TIMP-2 were counteracted by the three LAB strains. On the other hand, *L. plantarum* and *L. brevis* had a better effect on reducing Col1A1 than *W. cibaria*. Moreover, no significant differences in the TIMP-1 level were found. LABs and control exhibited similar levels to TGF-β.

## LAB attenuates TGF-β induced proinflammatory cytokine in LX-2 cells

This study examined whether LAB strains alter TGF-β induced proinflammatory cytokine/chemokine expressions *in vitro*. Fig 4 presents the RT-PCR results. Pre-incubating the LX-2 cells with the LAB strains produced lower mRNA levels of all cytokines (IL-6, CXCL8, CCL-2, IL-1β, and TNF-α) and TRAF6. On the other hand, these LAB-mediated reductions were strain-dependent. The IL-6, CXCL8, and TNF-α expression levels were diminished significantly by the LAB strains except for *W. cibaria*. In contrast, the *L. brevis* strain did increase the levels of IL-1β and CCL2, whereas the other strains, including *W. cibaria*, decreased the IL-1β and CCL2 levels. Among the cytokines, IL-6 was the only cytokine that was reduced by all the LAB strains.

## Attenuation of TGF-β signaling pathway

LAB could reduce the levels of both profibrogenic and proinflammatory markers *in vitro*. Therefore, LX-2 cells were treated with TGF-β for 120 minutes after the cells were pre-incubated with the LAB strains to determine how LAB strains reduce both these markers. The levels of profibrogenic (α-SMA and Col1A1) and TGF-β signaling-associated proteins (p-SMAD2/3 and SMAD7) were analyzed by western blot. A TGF-β treatment in the absence of the LAB strains upregulated the levels of α-SMA and CollA1 and the phosphorylation of p-SMAD2/3 in LX-2 cells. Pretreatments with the LAB strains could modulate the TGF-β induced proteins (Fig 5). Stimulation of cells with the LAB strains in the presence of TGF-β reduced the levels of α-SMA and Col1A1 in LX-2 cells significantly. In addition, the level of SMAD2/3 was reduced by *L. plantarum* and *L. brevis*, while the SMAD7 level was increased, indicating that the LAB strains exhibited anti-fibrogenic activity by modulating the SMAD proteins.

## Attenuation of MAPKs and NF-κB pathways

The levels of MAPKs and NF-κB proteins in LX-2 cells were next analyzed to confirm the mediation of the anti-inflammatory effect of LAB via modulating these two pathways *in vitro*. As shown in Fig 6, pretreatments with the LAB strains could modulate the phosphorylation of ERK, p38, and IκB-α in LX-2 cells before TGF-β stimulation. *L. plantarum* and *L. brevis* decreased the level of p-ERK and p-IκB-α in response to TGF-β in LX-2 cells. On the other hand, only *L. plantarum* produced a significant reduction in the level of MAPK-p38 phosphorylation as compared to TGF-β.

## LAB pretreatment attenuates TGF-β induced autophagy in LX-2 cells

LX-2 cells exposed to TGF-β increased the levels of the autophagy markers, such as LC3I/II, and ATG5, whereas the p62 level decreased. The LAB strains pretreatment followed by TGF-β stimulation modulated the levels of the autophagy markers. *L. plantarum* and *L. brevis* had lower levels of LC3I/II compared to TGF-β (Fig 7). The levels of p62 were not increased significantly when the cells were pretreated with LAB before TGF-β stimulation. In contrast, LAB had a remarkable effect on the level of TGF-β-induced ATG5, but the results were similar to the control and TGF-β alone. In addition, the LX-2 cells treated with TGF-β showed a decrease in the p-AKT and p-mTOR levels. The LAB pretreatment had higher levels of p-AKT and p-mTOR except for *L. brevis*.

## LAB pretreatment modulates TGF-β-induced apoptosis in LX-2 cells

TGF-β exposure resulted in a decrease in the level of apoptosis marker Bcl2 and an increase in levels of Bax and pro-caspase 3. On the other hand, these levels were modulated significantly

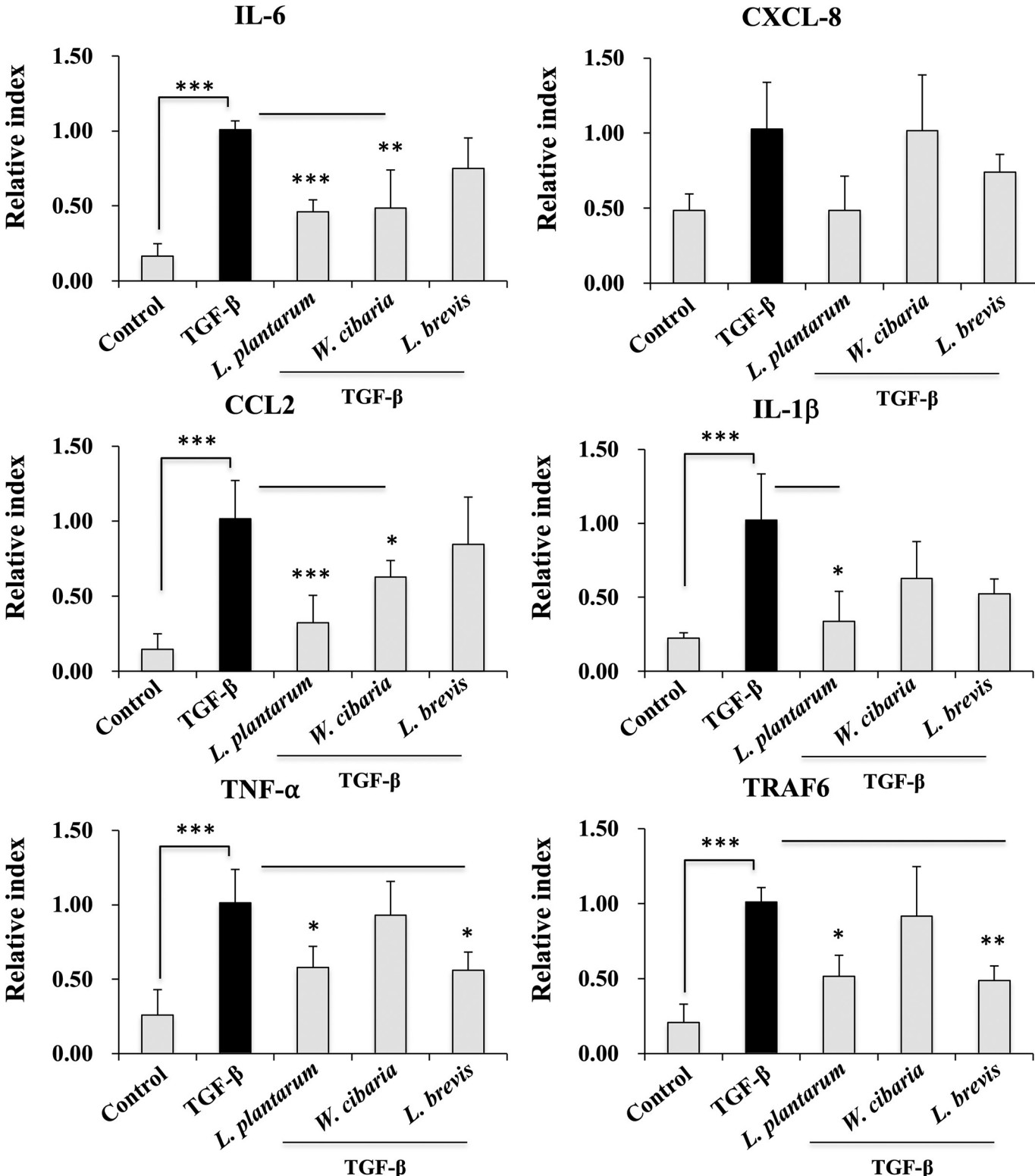

**Fig 4. Effect of LAB on TGF-β induced proinflammatory cytokine/chemokine expression in LX-2 cells.** Confluence LX-2 cells were treated with LAB for 48 h, and stimulated with TGF-β for 24 h. The IL-6, CXCL-8, CCL2, IL-1β, TNF-α, and TRAF6 expression levels were determined by RT-PCR. The LX-2 cells, either with medium or TGF-β alone, were used as the negative and positive control. Two independent experiments (n = 3) were performed, and the average values (mean + S.D) are shown. $^*p < 0.05$, $^{**}p < 0.01$, $^{***}p < 0.001$.

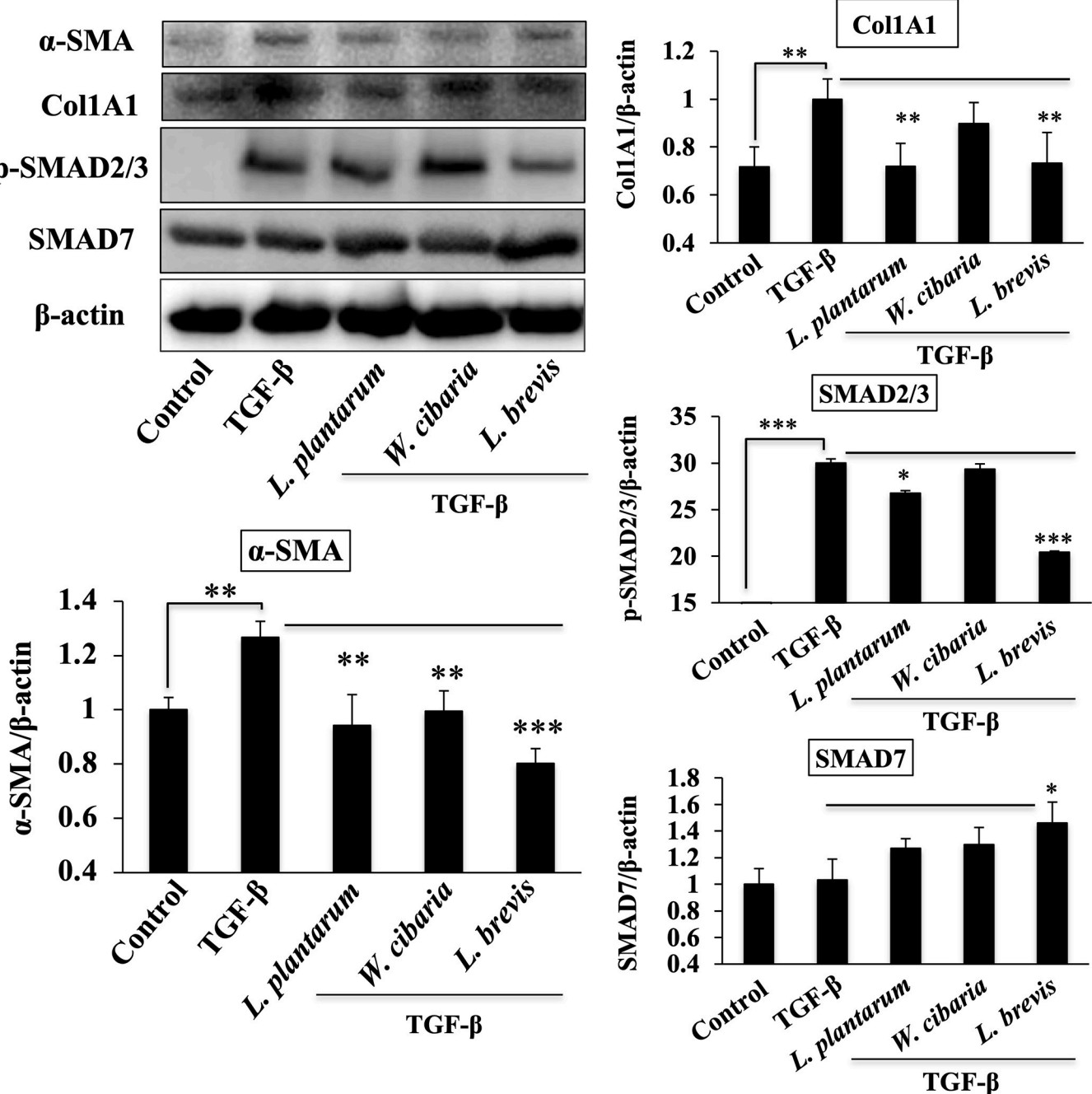

**Fig 5. LAB strains modulate TGF-β induced fibrogenic markers and SMAD-dependent TGF-β signaling in LX-2 cells.** Confluence LX-2 cells were treated with LABs for 48 h, and stimulated with TGF-β for 120 minutes. Western blot was performed to determine the level of α-SMA, Col1A1, p-SMAD2/3, and SMAD7 at the indicated time point. Image J software was used to determine the intensities of protein bands. The bar graphs are the average of two different experiments (mean + S.D). $^*p < 0.05$, $^{**}p < 0.01$, $^{***}p < 0.001$.

when the LX-2 cells had been pretreated with the LAB strains (Fig 8). All LAB upregulated the Bcl2 level, but the levels of Bax and caspase-3 were reduced only by *L. plantarum* and *W. cibaria* in response to TGF-β in the LX-2 cells. *L. brevis* had no remarkable effect on both Bax and caspase-3 that were relatively similar to TGF-β alone.

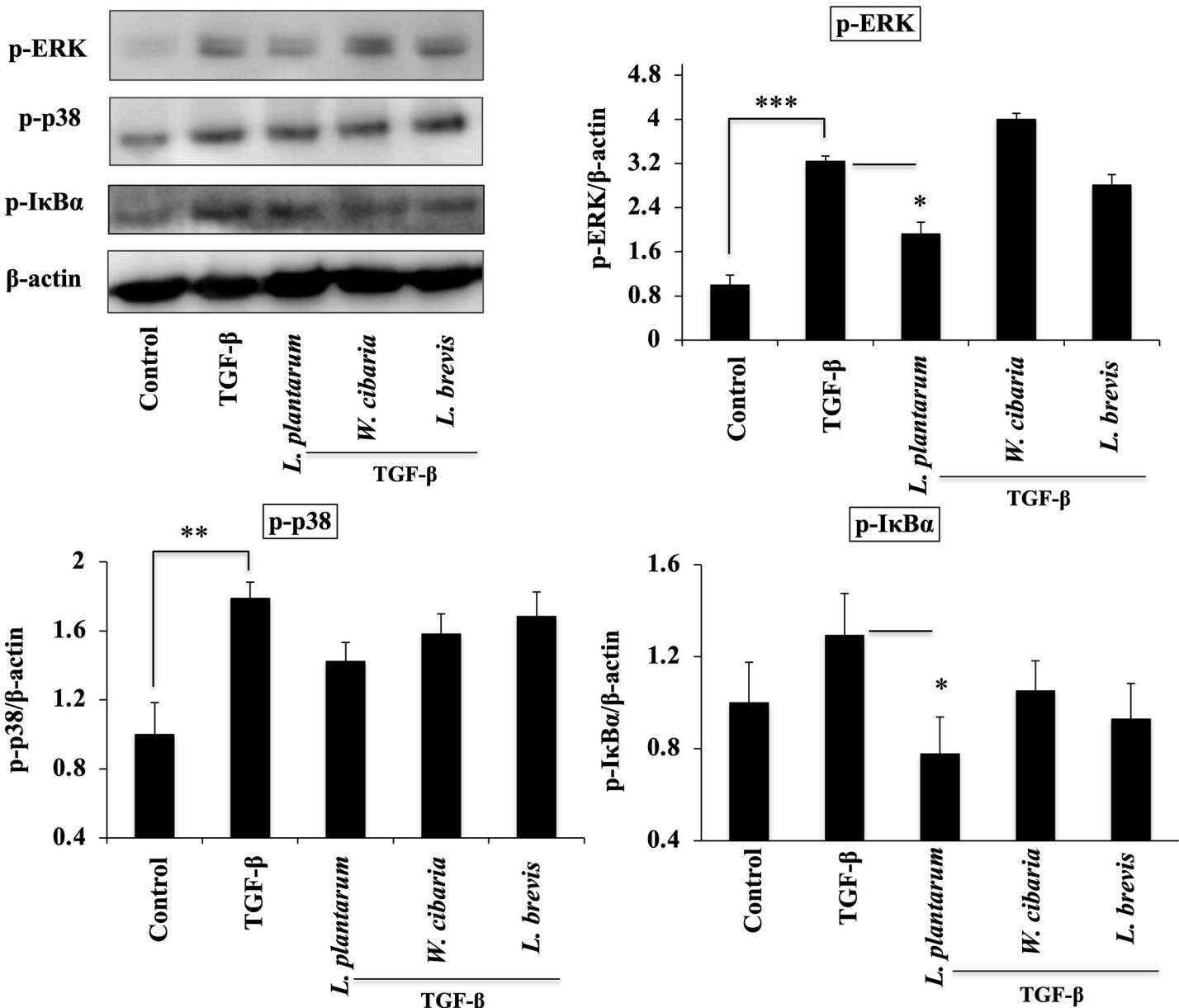

**Fig 6. LAB strains modulate SMAD-independent TGF-β signaling in LX-2 cells.** Confluence LX-2 cells were treated with LABs for 48 h, and stimulated with TGF-β for 120 minutes. Western blot was performed to determine the phosphorylation of ERK, p38, and IκBα at the indicated time point. The Image J software was used to determine the intensities of protein bands. The bar graphs are the average of two different experiments (mean + S.D). $^{*}p < 0.05$, $^{**}p < 0.01$, $^{***}p < 0.001$.

## Discussion

Several factors, including chronic alcoholic consumption, diet, and viral hepatitis, have been reported to induce hepatic injury or inflammation that is sustained by the production of a series of cytokines/chemokines and the infiltration of inflammatory cells into the injured site, resulting in the progression of liver diseases, including fibrosis [18]. Several clinical and animal studies have evaluated the possibility of ameliorating hepatic inflammation using many strategies. Probiotics are considered a promising factor that has better effects on ameliorating the

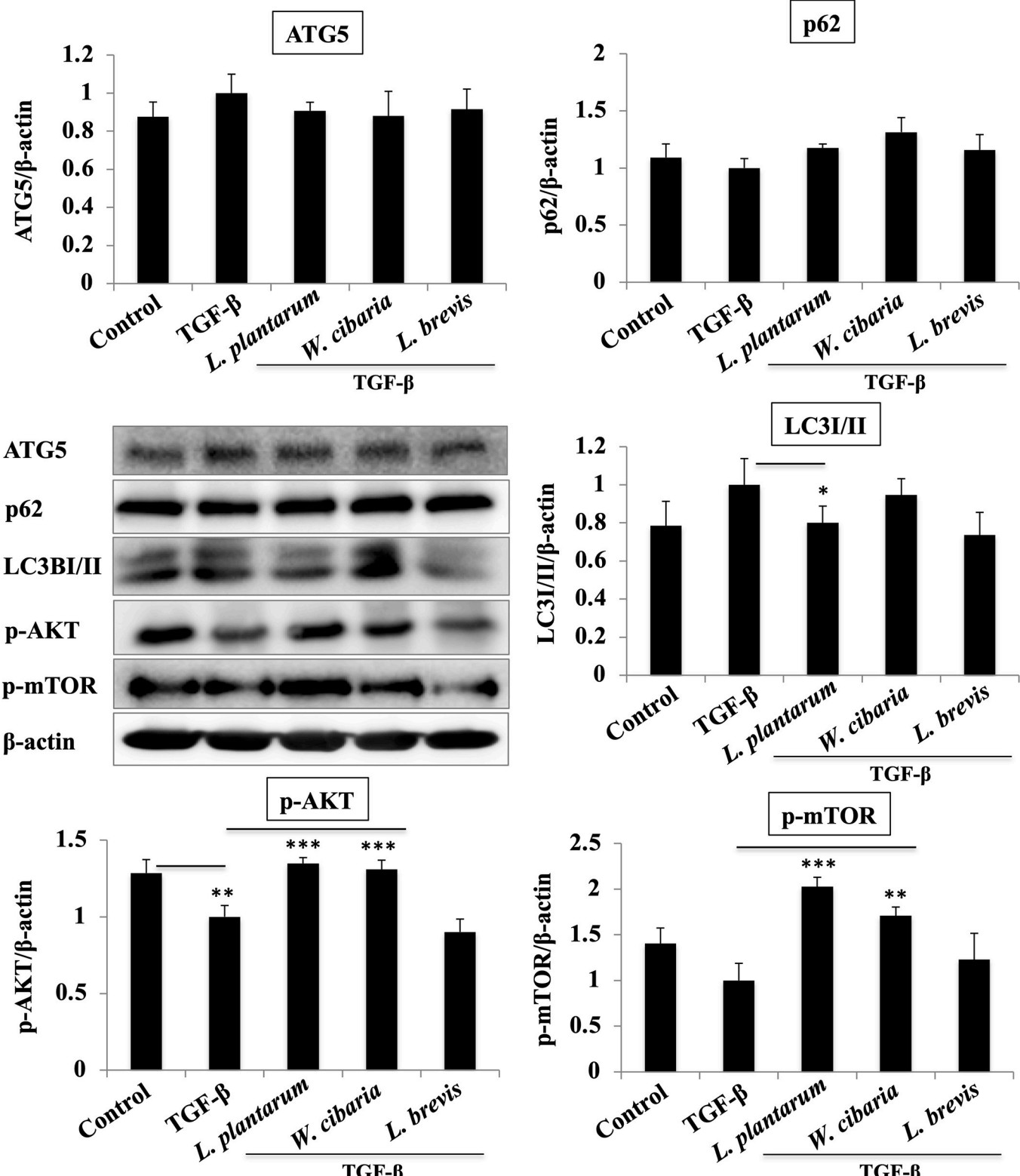

**Fig 7. LAB strains modulate the autophagy-associated markers in LX-2, in response to TGF-β.** Confluence LX-2 cells were treated with LABs for 48 h, and stimulated with TGF-β for 120 minutes. Western blot was performed to determine the level of ATG5, p62, LC3I/II, and p-AKT and p-mTOR at the indicated time point. Image J software was used to determine the intensities of the protein bands. The bar graphs are the average of two different experiments (mean +_S. D). $^{*}p < 0.05$, $^{**}p < 0.01$, $^{***}p < 0.001$.

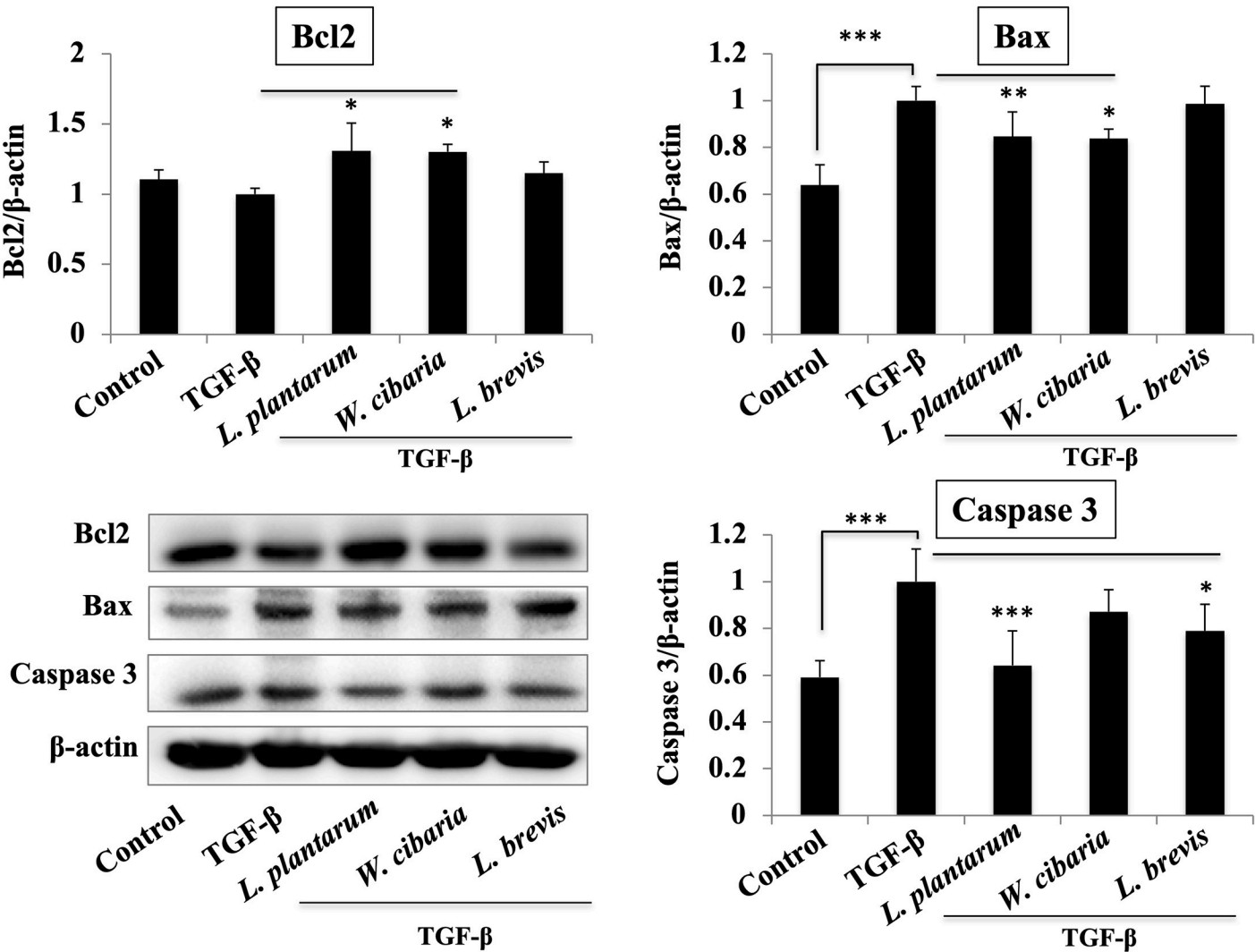

**Fig 8. LAB strains modulate the apoptosis-associated markers in LX-2 in response to TGF-β.** Confluence LX-2 cells were treated with LABs for 48 h, then stimulated with TGF-β for 120 minutes. Western blot was performed to determine the level of Bcl-2, Bax, and Caspase 3 at the indicated time point. The Image J software was used to determine the intensities of protein bands. The bar graphs are the average of two different experiments (mean + S.D.). $^*p$ <0.05, $^{**}$ $p$ <0.01, $^{***}p$ <0.001.

pathophysiological abnormalities of liver diseases without having systemic effects [7, 23, 24], but none of the groups examined the probiotic effects against fibrogenesis *in vitro*.

This study showed that pretreatment with the LAB strains produced a significant decrease in the level of pro-fibrogenic markers induced by TGF-β *in vitro*. TGF-β is a key pro-fibrogenic cytokine involved in the pathogenesis of liver fibrosis. Its overexpression or production activates HSCs and induces the extracellular matrix (ECM) synthesis in the liver [25, 26]. The presence of TGF-β induced collagen deposition in an *in vivo* mice model [27] and in vitro human HSCs [28]. Thus, TGF-β was used to induce HSCs to upregulate the production of collagen and other fibrogenic markers *in vitro*. Similar to other studies, TGF-β stimulation induced the secretion and deposition of collagen in LX-2 cells. Interestingly, exposure to LAB strains, such as *L. plantarum* and *L. brevis*, reduced the level of collagen deposition in LX-2 cells. Park et al. [21] reported that TGF-β stimulation upregulated the levels of pro-fibrogenic makers, such as α-SMA, fibronectin, MMP2, MMP-9, TIMP1, and TIMP2 in LX-2 cells.

Similar results were found with TGF-β in that a treatment increased the mRNA levels of the profibrogenic markers in HSCs. On the other hand, a LAB strain pretreatment significantly decreased the TGF-β-induced expression of Col1A1 and other profibrogenic markers, including α-SMA in LX-2 cells. α-SMA is a potent marker of liver fibrosis, which is produced by myofibroblasts and with collagen in the liver [29]. Myofibroblasts are a source of ECM overproduction in liver fibrosis. Thus, α-SMA-positive myofibroblasts were found increasingly in liver fibrous scars [29]. In addition to the pro-fibrogenic markers, TGF-β stimulation upregulated the expression of proinflammatory cytokines/chemokines in LX-2 cells. TGF-β induced IL-6, CXCL-8, CCL2, IL-1β, and TNF-α were reduced significantly by the LAB strains. In addition, TGF-β mediates signals via the non-SMAD pathways. The non-SAMD pathways first activate TNF receptor-associated factor 6 (TRAF6) to mediate inflammatory signaling and produce robust proinflammatory and pro-fibrogenic markers in liver cells. TRAF6 is a ubiquitin E3 ligase that interacts with the TGF-β receptor and aggravates hepatic inflammation and fibrosis by increasing the Lysine63-linked polyubiquitination of TGF-β-associated kinase 1 and apoptosis signal-regulating kinase 1 [30, 31]. In these results, pretreatment with the LAB strains reduced TGF-β-induced TRAF6 expression in LX-2 cells.

TGF-β stimulation not only induced the mRNA levels Col1A1 and α-SMA, but it also upregulated the protein levels in LX-2 cells. As stated earlier, TGF-β is a potent cytokine that activates HSCs and differentiates to collagen-producing myofibroblasts via an interaction with the TGF-β receptors [16]. The activation of TGF-β signaling decreases the bone morphogenetic protein and activin membrane-bound inhibitor (BAMBI), SMAD7, and increases the SMAD2/3/4 proteins, resulting in the excessive production of ECM and the progression of liver fibrosis [15, 17]. The SMAD pathway is responsible for the deposition of ECM and liver fibrosis. An in-vitro study also reported that HSCs-T6 exposed to TGF-β increased the phosphorylation and translocation of SMAD2/3 and SMAD4 to the nucleus [27]. In these results, TGF-β upregulated SMAD2/3, which are downstream of TGF-β mediating fibrous signaling in LX-2 cells. The LAB strains could modulate the TGF-β-induced SMAD2/3 and SMAD7 in LX-2 cells. TGF-β signaling activates the SMAD pathway, as well as some other signaling pathways, such as MAPKs, NF-κB, and AKT pathways, which are non-SMAD-dependent TGF-β signaling [18]. Both MAPKs and NF-κB are involved in modulating the physiological and pathophysiological responses *in vitro* and *in vivo*. LAB modulated TGF-β-induced MAPKs-ERK, p38, and IκBα levels in LX-2 cells. Hsu et al. [24] reported similar results in the hepatic injury mice model. Overall, these results suggest that LAB attenuates TGF-β induced pro-fibrogenic signaling by modulating both SMAD-dependent and SMAD-independent pathways in LX-2 cells.

Autophagy is a complex and tightly regulated process involved in the pathogenesis of liver fibrosis and liver cirrhosis [32, 33]. Autophagy-released lipids have been reported to induce hepatic fibrogenesis by activating HSCs in human and animal tissues [34]. The process of autophagy is like a double-edged sword and plays a role in cell survival, metabolism, and others, but its over-activation induces cell death and injury [35, 36]. The activation of autophagy exhibits its pro-fibrogenic effects via the activation of HSCs [34], while its inhibition attenuates liver fibrosis through the reduction of HSCs activation [37, 38]. The inhibition of autophagy activation may be a regulatory mechanism for the progression of liver fibrosis. Therefore, this study analyzed whether any changes occur in the process of autophagy after TGF-β stimulation *in vitro*. TGF-β stimulation induced autophagy-associated factors, such as LC3I/II, ATG5, and p62, in LX-2 cells. LC3I/II is an important marker of autophagic activation that is recruited from the cytoplasm to autophagosomes upon autophagic activation. Pretreatment with the LAB strains (*L. plantarum* and *L. brevis*) reduced TGF-β-induced LC3I/II but did not alter the level of p62 in LX-2 cells. In addition, TGF-β stimulation reduced the phosphorylation of AKT

and a mammalian target of rapamycin (mTOR) in LX-2 cells. Both AKT and mTOR are the central regulators of metabolisms and physiology and play important roles in the regulation of autophagy [39]. Exposure to the LAB strains (*L. plantarum* and *W. cibaria*) increased the AKT and mTOR levels in response to TGF-β in LX-2 cells, indicating that the LAB strains could modulate TGF-β-induced autophagy through modulation of the AKT and mTOR pathways.

Autophagy is closely associated with apoptosis, and their mutual inhibition shares similar signaling pathways and stimulating factors [40]. Apoptosis is a cell death process induced by several factors, such as oxidative stress, viruses, UV light, drugs, and chemicals. The apoptosis process is controlled by several genes, but Bcl2, Bax, and caspase 3 are the most common and reliable apoptosis markers [41]. Bcl2 and Bax have separate roles in apoptosis; Bcl2 is involved in the anti-apoptotic process, while Bax accelerates the cell apoptotic process [41]. Some studies reported that TGF-β induces autophagy and apoptosis simultaneously when mammary epithelial cells are exposed to TGF-β [42]. This study showed that TGF-β stimulation reduced the level of Bcl2 and increased the Bax and Caspase 3 levels in LX-2 cells. The LAB strains modulated these TGF-β-induced apoptosis makers. The LAB (*L. plantarum* and *W. cibaria*) pretreatment increased the level of Bcl2, while diminished the levels of Bax and caspase 3 in LX-2 cells, suggesting that the LAB strains could modulate TGF-β induced apoptosis *in vitro*.

## Conclusion

TGF-β can induce collagen deposition, profibrogenic markers, and anti-inflammatory cytokines in human LX-2 cells by activating the TGF-β signaling, autophagy, and apoptotic pathways. Pretreatment with the LAB strains had beneficial effects on LX-2 cells in response to TGF-β. In addition, the LAB strains could reduce collagen deposition, pro-fibrogenic factors, and inflammatory cytokines/chemokines in LX-2 cells. The beneficial effects of the LAB strains on LX-2 cells are probably due to the modulation of TGF-β induced pro-fibrogenic signaling, autophagy, and apoptotic pathways. On the other hand, further *in vivo* studies will be needed to confirm these results.

## Supporting information

**S1 Fig. Cytotoxicity of LAB strains on LX-2 cells.** LX-2 cells incubated with LAB strains for 48 h and then analyzed the cytotoxicity of LAB strains using cytotoxicity assay kit. The results are shown an average of two different experiments (mean + S.D).
(TIFF)

**S1 Table. List of primers used for RT-PCR.**
(DOCX)

**S1 Raw images. Original blots for Fig 5–8.**
(ZIP)

## Author Contributions

**Data curation:** Paulraj Kanmani.

**Funding acquisition:** Paulraj Kanmani, Hojun Kim.

**Project administration:** Hojun Kim.

**Resources:** Hojun Kim.

**Supervision:** Hojun Kim.

**Visualization:** Paulraj Kanmani.

**Writing – original draft:** Paulraj Kanmani.

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
