## [Decision Letter · Decision Letter 0]

6 Jul 2021

PONE-D-21-18946

Immunobiotics counteract the expression of hepatic profibrotic genes via the attenuation of TGF-b/SMAD signaling and autophagy in hepatic stellate cells

PLOS ONE

Dear Dr. Kim,

Thank you for submitting your manuscript to PLOS ONE. After careful consideration, we feel that it has merit but does not fully meet PLOS ONE’s publication criteria as it currently stands. Therefore, we invite you to submit a revised version of the manuscript that addresses the points raised during the review process.

We have received mixed recommendation for your manuscript and both reviewers provided useful comments. Please address all the comments in your revised version.

We look forward to receiving your revised manuscript.

Kind regards,

Partha Mukhopadhyay, Ph.D.

Academic Editor

PLOS ONE

Journal Requirements:

2. PLOS ONE now requires that submissions reporting blots or gels include original, uncropped blot/gel image data as a supplement or in a public repository. This is in addition to complying with our image preparation guidelines described at https://journals.plos.org/plosone/s/figures#loc-blot-and-gel-reporting-requirements. These requirements apply both to the main figures and to cropped blot/gel images included in Supporting Information. As such, we ask you to provide images of the full, uncut blots (not the cut strips) as a supplemental file.

Reviewers' comments:

Reviewer's Responses to Questions

**Comments to the Author**

1. Is the manuscript technically sound, and do the data support the conclusions?

Reviewer #1: Yes

Reviewer #2: Yes

2. Has the statistical analysis been performed appropriately and rigorously? 

Reviewer #1: Yes

Reviewer #2: Yes

3. Have the authors made all data underlying the findings in their manuscript fully available?

Reviewer #1: Yes

Reviewer #2: No

4. Is the manuscript presented in an intelligible fashion and written in standard English?

Reviewer #1: Yes

Reviewer #2: Yes

5. Review Comments to the Author

Reviewer #1: This manuscript suggested the role of lactic acid bacteria (LAB) in the attenuation of HSC activation. The treatment of LAB attenuates TGF-b/SMAD and autophagy signaling and eventually, LABs might be a tentative candidate to attenuate the progression of liver fibrosis. The authors used human HSC cell line (LX-2) to evaluate anti-fibrotic effects of selected probiotics against TGF-b and molecular mechanisms. In this aspect, this manuscript tried to identify the underlying mechanisms to explorer the novel role of probiotics in the attenuation of HSC activations. The concept in this manuscript is very straight forward, however, some critical concerns must be addressed.

<concerns>

1. The authors use 3 LAB stains (L.plantarum, W.cibaria, and L.brevis) throughout this study. However, there was no information of the probiotics concentrations/amount (such as CFU/g). It must be provided.

2. Before this study, the evaluation of cytotoxicity is very important. The authors mentioned the cytotoxicity of LAB was done (materials and methods). However, there was not further information about those cytotoxicity. The cytotoxicity results must be shown for probiotic concentration/amount.

3. The authors used heat-killed LABs (materials and methods). After heat-killing of LABs, the evaluation of cytotoxicity is meaningful? (it looks like LABs were all inactivated). Explain more.

4. The authors used in vitro experiments using LX-2 and it seems that LABs had certain protective effects in the progression of HSC activations. Albeit those in vitro data, it is very difficult/cautious to draw a conclusion that LABs have a beneficial role in the liver fibrosis. The process of liver fibrosis is very complex and many different cells are involved in. To draw the results, in vivo mouse model must be included. Hepatic fibrosis with/without pretreatment of LABs (maybe via oral administration) must be done for this conclusion.

5. In figure1, the order of X-axis needs to be changed according to other data set. Control -> TGF-b -> L.plantarum -> W.cibaria -> L.brevis

6. The data set is about HSC activation (figure3) and proinflammatory cytokines (figure4). It is recommended combined together figure3 and figure 4 to a “one figure”

7. Up to figure4, the authors used pretreatment of LAB (48h) and TGF-b treatment for 24 hours. However, TGF-b treatment was only 120 minutes in figure 5,6,7. Was there any specific reasons to change time frame?

8. In figure 6 and 7 western blotting data, the authors used p-p38,p-ERK, IkB alpha and p-AKT. In this case, total form of those proteins (total ERK, total p38…) must be added.

9. The title is immunobiotics counteract the expression of hepatic profibrotic genes via the attenuation of TGF-b/SMAD signaling and autophagy in HSC. However, there was no further use “immunobiotics” in the manuscript any longer. Rather than immunobiotics, probiotics were used in the manuscript. For the consistence the manuscript, the title needs to changed.

10. In abstract, there was no full name of LAB strains. Full name of LAB needs to be provided to first readers.

11. In manuscript, the authors use HSC cells everywhere. The abbreviation of HSC is hepatic stellate cells. Therefore, HSC cells needs to be changed to HSC (singular) or HSCs (pleural).</concerns>

Reviewer #2: The authors tapped into the interesting topic of hepatic fibrosis and attempted to explore the mechanisms through a specific immunoregulatory scenario related to probiotics.

The exploration was done fully in vitro, and human immortalized hepatic stellate cell line LX-2 was selected as a platform for the experiments. The treatment of probiotic strains to LX-2 cells and subsequent stimulus by TGF-beta in cell culture environment was the highlight of the experimental model and served as the basis of authors’ narratives. A suite of mRNA biomarkers was tested to support the hypothesis. Furthermore, SMAD signaling and autophagy was explored to extract mechanistic evidence in supporting the author’s hypothesis of probiotics ameliorating TGF-beta induced liver fibrosis via regulating hepatic stellate cells.

With mounting results, the authors concluded that stellate cell line LX-2 treated with lactic acid bacteria strains showed reduced collagen deposition, pro-fibrogenic factors and inflammatory cytokines, and that LAB strains are beneficial to LX-2 cells due to modulatory effects in TGF-beta induced profibrogenic signalling and autophagy pathways.

I have a few suggestions in minor editing and some questions for the authors:

1. Please include the list of RT-PCR primers in supplement materials.

2. Can authors elaborate in more details regarding LX-2 cells’ stimulation by LAB strains: are LX-2 cells stimulated with heat-killed LAB suspension? Which media was used to introduce LAB to LX-2 cell culture and how was control groups treated respectively?

3. Please clearly indicate the meaning of “Relative Index” in graphs expressing RT-PCR results of mRNA relative expressions. Are you using 2^(-ddCt) method? Are final results graphed in linear scale or logarithmic scale?

4. Please correct Figure 3 COL1A-1 graph title to make upper/lower case consistent.

5. The use of antibiotics in initial culture of LX-2 might invite the question of whether it affects the baseline LX-2 response to TGF-beta and expression profile of mRNA biomarkers. Can author please address this?

6. What were the TIMP-1 and TIMP-1 mRNA expression levels relative to beta-Actin in RT-PCR assays? LX-2 cell lines have dampened TIMP-1 protein secretion as opposed to LX-1 or primary HSC, which might contribute to the non-differentiating result of TIMP-1 mRNA expression in the experiment. Are TIMP-1/TIMP-2 protein measured in cell culture supernatant?

7. The selection of LAB strains requires some explaining, as the three strains yielded quite inconsistent results across the board, with L. plantarum arguably produced best results that suited authors’ hypothesis. The characteristics of the 3 strains

8. Admittedly this study solely focused on in vitro explorations, the significance of the results remains in question with consideration of translation to in vivo scenario, which liver fibrosis is. With the way (LX-2) cells are treated with LAB strains in vitro, given the structural features of liver sinusoids, portal/hepatic vasculature and GI tracts, how is this study relatable to the potential therapeutic application of LAB in combating liver fibrosis?

6. PLOS authors have the option to publish the peer review history of their article (what does this mean?). If published, this will include your full peer review and any attached files.

Reviewer #1: No

Reviewer #2: No

---

## [Author Response · Author response to Decision Letter 0]

26 Jul 2021

Response to the Reviewer

Reviewer #1: 

This manuscript suggested the role of lactic acid bacteria (LAB) in the attenuation of HSC activation. The treatment of LAB attenuates TGF-b/SMAD and autophagy signaling and eventually, LABs might be a tentative candidate to attenuate the progression of liver fibrosis. The authors used human HSC cell line (LX-2) to evaluate anti-fibrotic effects of selected probiotics against TGF-b and molecular mechanisms. In this aspect, this manuscript tried to identify the underlying mechanisms to explorer the novel role of probiotics in the attenuation of HSC activations. The concept in this manuscript is very straight forward, however, some critical concerns must be addressed.

1. The authors use 3 LAB stains (L.plantarum, W.cibaria, and L.brevis) throughout this study. However, there was no information of the probiotics concentrations/amount (such as CFU/g). It must be provided.

Answer: First, we would like to thank for your valuable review and your kind suggestions. We have mentioned in the methods section (Page 5-8). 

2. Before this study, the evaluation of cytotoxicity is very important. The authors mentioned the cytotoxicity of LAB was done (materials and methods). However, there was not further information about that cytotoxicity. The cytotoxicity results must be shown for probiotic concentration/amount.

Answer: Thanks, we included cytotoxicity of probiotic strains in the suppl figure 1. 

3. The authors used heat-killed LABs (materials and methods). After heat-killing of LABs, the evaluation of cytotoxicity is meaningful? (it looks like LABs were all inactivated). Explain more.

Answer: Thanks; we agree with the reviewer. Heat-killed LABs have no cytotoxic effect on the cells; however, we just want to confirm by the experiments. 

4. The authors used in vitro experiments using LX-2 and it seems that LABs had certain protective effects in the progression of HSC activations. Albeit those in vitro data, it is very difficult/cautious to draw a conclusion that LABs have a beneficial role in the liver fibrosis. The process of liver fibrosis is very complex and many different cells are involved in. To draw the results, in vivo mouse model must be included. Hepatic fibrosis with/without pretreatment of LABs (maybe via oral administration) must be done for this conclusion.

Answer: Thanks; Yes, many cells (KC and HSC) participate role in the hepatic fibrosis. However, most cell types actively involved in the induction of hepatic inflammation. HSC is a main cell type that actively respond to TGF-beta and activate into extracellular matrix (ECM) producing myofibroblasts, resulting in development of liver fibrosis. 

In addition, we have confirmed the anti-fibrogenic effect of probiotics by in vivo study. We found that oral administration (Treatment and Prevention) of probiotics (Live and Heat-Killed) effectively reduced diet (MCD) induced NASH, liver fibrogenesis, endotoxic, modulated T cells (Th1, Th17, Th2, Treg) and gut microbial composition in the mice. It’s very big study, the results of in vivo study are in under review (Kanmani et al., 2021). Therefore, we are not able to show the data here, we apologize for the inconvenience. 

5. In figure1, the order of X-axis needs to be changed according to other data set. Control -> TGF-b -> L.plantarum -> W.cibaria -> L.brevis

Answer: Thanks; we have reordered the data set according to the reviewer suggestion. 

6. The data set is about HSC activation (figure3) and proinflammatory cytokines (figure4). It is recommended combined together figure3 and figure 4 to a “one figure”

Answer: Thanks; the Figure 3 (Profibrogenic markers) and Figure 4 (Pro-inflammatory markers) are two different markers, therefore we showed separately to readers understand well. Combine both may confuse readers, it’s our thoughts, if reviewer still wants to combine both, we will do it next time.

7. Up to figure4, the authors used pretreatment of LAB (48h) and TGF-b treatment for 24 hours. However, TGF-b treatment was only 120 minutes in figure 5,6,7. Was there any specific reasons to change time frame?

Answer: Thanks; there is no specific reasons, however mRNA level of cytokines is usually analyzed in long term treatment, but the protein levels analyzed within a relatively short period time. 

8. In figure 6 and 7 western blotting data, the authors used p-p38,p-ERK, IkB alpha and p-AKT. In this case, total form of those proteins (total ERK, total p38…) must be added.

Answer: Thanks; we apologize for the inconveniences, the first author already moved to another place. In addition, beta-actin can be used to as alternative to compare the phosphorylated form proteins. 

9. The title is immunobiotics counteract the expression of hepatic profibrotic genes via the attenuation of TGF-b/SMAD signaling and autophagy in HSC. However, there was no further use “immunobiotics” in the manuscript any longer. Rather than immunobiotics, probiotics were used in the manuscript. For the consistence the manuscript, the title needs to changed.

Answer: Thanks, we have changed the title according to the reviewer suggestion. 

10. In abstract, there was no full name of LAB strains. Full name of LAB needs to be provided to first readers.

Answer: Thanks, we have included full names of LAB strains in the abstract section according to the reviewer suggestion (Page 2, lines 5-6). 

11. In manuscript, the authors use HSC cells everywhere. The abbreviation of HSC is hepatic stellate cells. Therefore, HSC cells needs to be changed to HSC (singular) or HSCs (pleural).

Answer: Thanks; we have modified HSC cells to HSCs according to the reviewer suggestion. 

 

Reviewer #2: 

The authors tapped into the interesting topic of hepatic fibrosis and attempted to explore the mechanisms through a specific immunoregulatory scenario related to probiotics.

The exploration was done fully in vitro, and human immortalized hepatic stellate cell line LX-2 was selected as a platform for the experiments. The treatment of probiotic strains to LX-2 cells and subsequent stimulus by TGF-beta in cell culture environment was the highlight of the experimental model and served as the basis of authors’ narratives. A suite of mRNA biomarkers was tested to support the hypothesis. Furthermore, SMAD signaling and autophagy was explored to extract mechanistic evidence in supporting the author’s hypothesis of probiotics ameliorating TGF-beta induced liver fibrosis via regulating hepatic stellate cells.

With mounting results, the authors concluded that stellate cell line LX-2 treated with lactic acid bacteria strains showed reduced collagen deposition, pro-fibrogenic factors and inflammatory cytokines, and that LAB strains are beneficial to LX-2 cells due to modulatory effects in TGF-beta induced profibrogenic signalling and autophagy pathways.

I have a few suggestions in minor editing and some questions for the authors:

1. Please include the list of RT-PCR primers in supplement materials. 

Answer: First, we would like to thank for your valuable review and your kind suggestions. We have included the primer lists as supplement table 1. 

2. Can authors elaborate in more details regarding LX-2 cells’ stimulation by LAB strains: are LX-2 cells stimulated with heat-killed LAB suspension? Which media was used to introduce LAB to LX-2 cell culture and how was control groups treated respectively?

Answer: Thanks; LX-2 cells stimulated with heat-killed LAB strains. We first mentioned the preparation of LAB strains in method section (Lactic acid bacteria, Page no 5), therefore we further mentioned LX-2 cells stimulated with LAB strains. 

3. Please clearly indicate the meaning of “Relative Index” in graphs expressing RT-PCR results of mRNA relative expressions. Are you using 2^(-ddCt) method? Are final results graphed in linear scale or logarithmic scale?

Answer: Thanks; we showed results of RT-PCR with relative expression and linear scale. 

4. Please correct Figure 3 COL1A-1 graph title to make upper/lower case consistent.

Answer: Thanks; We have modified the graph title in Figure 3 according to the reviewer suggestion. 

5. The use of antibiotics in initial culture of LX-2 might invite the question of whether it affects the baseline LX-2 response to TGF-beta and expression profile of mRNA biomarkers. Can author please address this?.

Answer: Thanks; we used antibiotics to avoid the contaminations with bacteria or others. Addition of antibiotics won’t affect the capability of LX-2 cells to respond TGF-beta and express genes. 

6. What were the TIMP-1 and TIMP-1 mRNA expression levels relative to beta-Actin in RT-PCR assays? LX-2 cell lines have dampened TIMP-1 protein secretion as opposed to LX-1 or primary HSC, which might contribute to the non-differentiating result of TIMP-1 mRNA expression in the experiment. Are TIMP-1/TIMP-2 protein measured in cell culture supernatant?

Answer: Thanks; the results of RT-PCR for TIMP-1 and 2 were already showed in Figure 3. Through this study, we evaluate whether probiotics modulate TGF-beta induced TIMP1 and TIMP2 production in LX-2 cells. LX-2 cells increased the production of TIMP-1 in response to TGF-beta. Yes, we can analyze the TIMP levels in CFS by ELISA technique.

7. The selection of LAB strains requires some explaining, as the three strains yielded quite inconsistent results across the board, with L. plantarum arguably produced best results that suited authors’ hypothesis. The characteristics of the 3 strains.

Answer: Thanks; we selected these strains based on their anti-bacterial, anti-viral, and anti-inflammatory effects in vitro using human intestinal epithelial cells (Kanmani et al., 2018, 2020). We further evaluated their efficacy against liver fibrosis in vitro, overall, L. plantarum shows better results than other strains. We already confirmed their effect in vivo mice model, the results already submitted to the journals (Kanmani et al., 2021). 

8. Admittedly this study solely focused on in vitro explorations, the significance of the results remains in question with consideration of translation to in vivo scenario, which liver fibrosis is. With the way (LX-2) cells are treated with LAB strains in vitro, given the structural features of liver sinusoids, portal/hepatic vasculature and GI tracts, how is this study relatable to the potential therapeutic application of LAB in combating liver fibrosis?.

Answer: Thanks; We just confirmed effect of probiotics on modulation of TGF-beta (main factor leads pro-fibrogenic signaling) induced fibrogenic signaling in LX-2 cells. We have also confirmed the anti-fibrogenic effect of probiotics by in vivo study. We found that oral administration (Treatment and Prevention) of probiotics (Live and Heat-Killed) effectively reduced diet (MCD) induced NASH, liver fibrogenesis, endotoxic, modulated T cells (Th1, Th17, Th2, Treg) and gut microbial composition in the mice. It’s very big study, the results of in vivo study are in under review (Kanmani et al., 2021). Therefore, we are not able to show the data here, we apologize for the inconvenience.

---

## [Decision Letter · Decision Letter 1]

15 Nov 2021

PONE-D-21-18946R1Immunobiotics counteract the expression of hepatic profibrotic genes via the attenuation of TGF-b/SMAD signaling and autophagy in hepatic stellate cellsPLOS ONE

Dear Dr. Kim,

Thank you for submitting your manuscript to PLOS ONE. After careful consideration, we feel that it has merit but does not fully meet PLOS ONE’s publication criteria as it currently stands. Therefore, we invite you to submit a revised version of the manuscript that addresses the points raised during the review process.

We received positive feedback from both but bone of them raised a minor issue, which require revision. A quick editorial decision will be taken after receiving revised manuscript addressing that.

We look forward to receiving your revised manuscript.

Kind regards,

Partha Mukhopadhyay, Ph.D.

Academic Editor

PLOS ONE

Journal Requirements:

Reviewers' comments:

Reviewer's Responses to Questions

**Comments to the Author**

1. If the authors have adequately addressed your comments raised in a previous round of review and you feel that this manuscript is now acceptable for publication, you may indicate that here to bypass the “Comments to the Author” section, enter your conflict of interest statement in the “Confidential to Editor” section, and submit your "Accept" recommendation.

Reviewer #1: All comments have been addressed

Reviewer #2: All comments have been addressed

2. Is the manuscript technically sound, and do the data support the conclusions?

Reviewer #1: Yes

Reviewer #2: Yes

3. Has the statistical analysis been performed appropriately and rigorously? 

Reviewer #1: Yes

Reviewer #2: Yes

4. Have the authors made all data underlying the findings in their manuscript fully available?

Reviewer #1: Yes

Reviewer #2: Yes

5. Is the manuscript presented in an intelligible fashion and written in standard English?

Reviewer #1: Yes

Reviewer #2: Yes

6. Review Comments to the Author

Reviewer #1: In question 7, the authors analyzed mRNA level (48 hours) and protein expression (120 minutes). Regarding the central dogma, transcription will take place first and then translation. Could the authors consider this issue for analysis of mRNA and protein level to determine the time points?

In question 8, it is fully understandable that the first author is gone, however, it can not be a reason why the information is not showing in the figure. Please provide and repeat total form of those protein markers.

Reviewer #2: The authors addressed all my questions and made changes accordingly. important materials are added. There's no further questions from me.

7. PLOS authors have the option to publish the peer review history of their article (what does this mean?). If published, this will include your full peer review and any attached files.

Reviewer #1: No

Reviewer #2: No

---

## [Author Response · Author response to Decision Letter 1]

17 Dec 2021

Response to the Reviewer

Reviewer #1: 

In question 7, the authors analyzed mRNA level (48 hours) and protein expression (120 minutes). Regarding the central dogma, transcription will take place first and then translation. Could the authors consider this issue for analysis of mRNA and protein level to determine the time points?

Answer: First, we would like to thank for your valuable review and your suggestions. Several studies used similar time points for the mRNA level and protein level study in the TGF-beta mediated inflammation and fibrogenesis in vitro (Mei et al., 2011 PLoS ONE 6(12): e28915, Zhu et al., 2011, Mole Med rep 4: 505-509; Lynch et al., 2012; Carcinogenesis, 33:976-985; Linag et al., 2013, Biochem Pharmacol, 85: 1594-1602; Song et al., 2021; Exp Ther Med, 21(5): 420). We also published several papers with the similar time points. 

In question 8, it is fully understandable that the first author is gone, however, it can not be a reason why the information is not showing in the figure. Please provide and repeat total form of those protein markers.

Answer: Thank you for your concern. We express our sincerest apology and are sorry about inconveniences to the reviewer. However, we’d like to show that other several studies in renowned journals also used beta-actin as alternative to compare the phosphorylated form of proteins without total forms. (Zhang et al. J Biol Chem . 2015 Mar 27;290(13):8232-42., Csati et al/ J Headache Pain . 2015;16:99., Sizemore et al. J Biol Chem . 2002 Feb 8;277(6):3863-9.) We hope the reviewer to understand our situation that additional experiment is not be able to be performed and to accept our apologies. 

Reviewer #2: 

The authors addressed all my questions and made changes accordingly. important materials are added. There's no further questions from me.

Answer: Thank you so much.

---

## [Editor Report · Decision Letter 2]

5 Jan 2022

Probiotics counteract the expression of hepatic profibrotic genes via the attenuation of TGF-b/SMAD signaling and autophagy in hepatic stellate cells

PONE-D-21-18946R2

Dear Dr. Kim,

We’re pleased to inform you that your manuscript has been judged scientifically suitable for publication and will be formally accepted for publication once it meets all outstanding technical requirements.

Kind regards,

Partha Mukhopadhyay, Ph.D.

Section Editor

PLOS ONE
---

## [Editor Report · Acceptance letter]

10 Jan 2022

PONE-D-21-18946R2 

Probiotics counteract the expression of hepatic profibrotic genes via the attenuation of TGF-β/SMAD signaling and autophagy in hepatic stellate cells 

Dear Dr. Kim:

I'm pleased to inform you that your manuscript has been deemed suitable for publication in PLOS ONE. Congratulations! Your manuscript is now with our production department. 

Kind regards, 

on behalf of

Dr. Partha Mukhopadhyay 

Section Editor

PLOS ONE